artificial intelligence; neural networks; breast cancer; cancer diagnosis; histopathology

**Corresponding author:**
Arslaan Javaeed;
Email: arslaanjavaeed@yahoo.com

# Artificial intelligence in breast cancer diagnosis: A systematic literature review

Arslaan Javaeed ⬤ and Anna Schuh ⬤

Oncology, University of Oxford, UK

## Abstract

Breast cancer is the second leading cause of cancer-related deaths among women globally and the most prevalent cancer in women. Artificial intelligence (AI)-based frameworks have shown great promise in correctly classifying breast carcinomas, particularly those that may have been difficult to discern through routine microscopy. Additionally, mitotic number quantification utilizing AI technology is more accurate than manual counting. With its many advantages, such as improved accuracy, efficiency and consistency as shown in this literature review, AI has promise for significantly enhancing breast cancer diagnosis in the clinical world despite the paramount obstacles that must be addressed. Ongoing research and innovation are essential for overcoming these challenges and effectively harnessing AI's transformative potential in breast cancer detection and assessment.

## Impact statement

This article explores the impact of artificial intelligence (AI) on breast cancer (BC) diagnosis within the field of pathology. It examines several applications of AI in BC pathology and provides a succinct summary of the principal findings from multiple investigations. Incorporating AI with conventional pathology methods may enhance diagnostic accuracy and reduce preventable errors. Studies have shown the efficacy of AI in detecting invasive breast tumors by rapidly analyzing extensive whole-slide images. Advanced convolutional neural networks underpin these discoveries. AI-driven quantitative analysis has facilitated the assessment of an individual's hormonal status, which is crucial for determining the appropriate BC treatment. This is due to its facilitation of consensus among many observers regarding their findings. AI has the potential to become essential for assessing BC and quantifying mitotic cells, as it can accurately classify moderate-grade breast carcinomas. Furthermore, the utilization of AI for measuring mitotic numbers has proven to be more precise and sensitive than manual methods, resulting in enhanced predictive outcomes. In the context of triple-negative BC, to maximize the benefits of AI in BC pathology, it is essential to address issues such as the necessity for comprehensive annotations and the challenges of differentiation. Despite existing challenges, AI's numerous contributions to BC pathology indicate a promising future characterized by enhanced accuracy, efficiency and uniformity. It is imperative that we continue researching and developing novel approaches to address these challenges and fully harness AI's promise to revolutionize BC detection and assessment.

## Introduction

### History of artificial intelligence

Artificial intelligence (AI) refers to the utilization of technology and computers to imitate human-like cognitive processes and intelligent actions (Försch et al., 2021; Briganti, 2023). The history of computers can be traced back over 200 years, marked by several significant advancements. The exact year of the first computer's invention is uncertain, but it is commonly attributed to 1822, when Charles Babbage introduced a design for a functional computer on paper (Grzybowski et al., 2024). The history of AI spans several decades (Muthukrishnan et al., 2020), beginning in the 1950s with Alan Turing's research on the feasibility of intelligent machines, culminating in his landmark 1950 article (Turning, 1950; Kaul et al., 2020). In 1956, the term "AI" was coined by John McCarthy (Anyoha et al., 2017) during a conference at Dartmouth College, where the first AI program, "Logic Theorist," was introduced (Moor, 2006).

### Concepts in AI

AI applications in medicine have evolved significantly due to advancements in machine learning (ML) and deep learning (DL) (Lanzagorta-Ortega et al., 2022). AI-based models can assist in

diagnosing diseases, forecasting therapy responses and promoting preventive medicine (Kaul et al., 2020; Pettit et al., 2021).

### Machine learning

The term "ML" was first introduced by Arthur Samuel in 1959, with applications in medicine emerging in the 1980s and 1990s (Brown, 2021). As a subset of AI, ML uses algorithms to build models that can learn from data and help in segmentation, classification or make predictions (Jiang et al., 2022). It is classified into three categories: supervised, unsupervised and reinforcement learning (Hosny et al., 2018; Ono and Goto, 2022). Supervised learning trains models with input and output data to predict outcomes and unsupervised learning analyzes unannotated data to discover patterns without predefined results; similarly, reinforcement learning involves learning through interactions with an environment, receiving rewards or penalties based on actions taken (Jovel and Greiner, 2021; Lee et al., 2022 Jiang et al., 2022; Al-Hamadani et al., 2024).

### Deep learning

DL is a subset of ML using artificial neural networks (ANNs) with multiple layers (Sarker, 2021) effective in complex tasks and large datasets (Sidey-Gibbons and Sidey-Gibbons, 2019; Birhane et al., 2023). Neural networks are designed like biological neurological systems based on the fundamental unit perceptron or neuron, and usually comprise of an input, hidden and output layers (Kriegeskorte and Golan, 2019). Deep neural networks (DNNs) are advanced models with multiple hidden layers used in healthcare for medical imaging and diagnostics (Baji'et al., 2022; Egger et al., 2022). Some ANNs have no hidden layer while DNNs have multiple, enabling them to understand complex behaviors (Kufel et al., 2023). Convolutional neural networks (CNNs) are specifically designed for image recognition and classification tasks (Alajanbi et al., 2021). Recently, these models have shown great potential for accurate diagnoses such as diabetic retinopathy from retinal images (Ragab et al., 2022).

### AI in medicine

ML, a key technology in AI, is used across various medical specialties, including oncology, cardiology and neurology (Bitkina et al., 2023). AI applications include screening, diagnosis, treatment, drug development (Xu et al., 2023), genomic analysis, patient monitoring and wearable health technology (Shajari et al., 2023). Additionally, AI enhances doctor–patient interactions, enables remote therapy and manages large datasets (Shajari et al., 2023; Chen and Decary 2020; Basu et al. 2020). Integrating AI into healthcare can significantly improve the effectiveness, accuracy and personalization of medical diagnoses and treatments (Alowais et al., 2023).

### AI in cancer diagnosis

AI has the potential to significantly advance cancer diagnosis by using annotated medical data, advanced ML techniques and enhanced processing power (Sufyan et al., 2023; Alshuhri et al., 2024). These developments are expected to transform patient care by improving efficiency, accuracy and customization in diagnoses and treatments (Chen and Decary, 2020). Over the past decade, DL architectures have outperformed traditional ML methods in cancer diagnosis, effectively utilizing genomic and phenotype data for cancer classification and treatment (Miotto et al., 2018). Computer-aided detection and diagnosis (CADx) are playing important roles in clinical imaging and are expected to further improve (He et al., 2020; Jairam and Ha, 2022). AI technology has the potential to improve the accuracy of clinical image analysis for identifying cancer progression, aiding in the early detection and diagnosis, while medical imaging remains essential for early identification and monitoring of cancer (Suberi et al., 2017; Liu et al., 2020; Lathwal et al., 2020).

### AI in breast cancer pathology

Breast cancer (BC) is the most diagnosed cancer among women and the second leading cause of cancer-related deaths worldwide (Watkins et al., 2019), with approximately 2.3 million new cases and 6,85,000 fatalities reported in 2020 cancer (Hanna et al., 2017; Nardin et al., 2020). It represents 25% of all newly diagnosed cancer cases, and projections suggest an increase to nearly 2.96 million cases by 2040 (Sedeta et al., 2023). Accurate histopathological diagnosis is crucial, as it confirms the presence of tumor cells and helps classify the type and grade of cancer (Cardoso et al., 2019). Discrepancies in diagnoses can significantly affect treatment decisions, highlighting the need for precise pathologic assessments (Soliman et al., 2024).

Despite advancements in imaging-based diagnostics and therapies, the field of histopathology has been slow to digitize, beginning this process only about two decades ago (Hanna et al., 2019; Försch et al., 2021). Histopathological diagnosis has remained largely unchanged, still relying on microscopic evaluations by pathologists. This reliance can lead to errors, such as false positives or negatives, especially under stress (Morelli et al., 2013; Cohen et al., 2022). Studies have shown significant variability in pathologists' assessments, with a concordance rate of only 75.3% overall and a particularly low rate of 48% for ductal carcinoma in situ (DCIS) and atypical hyperplasia, indicating ambiguity in pathology interpretations (Elmore et al., 2015).

The use of AI in cancer diagnosis is essential for improving diagnostic processes and addressing the shortage of pathologists alongside the increasing number of cancer cases (Robboy et al., 2020). AI in pathology relies on whole-slide imaging (WSI) technology (Niazi et al., 2019; Ahn et al., 2023), which converts physical pathological slides into high-resolution digital images. These images are an abundant source of information – with sizes up to $1,00,000 \times 1,00,000$ pixels and are the first step in creating AI-assisted models (Mukhopadhyay et al., 2018; Niazi et al., 2019). WSI aids in the easy sharing and consultation of images, reducing interpretation errors and enhancing the analysis of complex cases (Hanna et al., 2019; Jones et al., 2015). Overall, WSI presents a promising alternative to conventional microscopic examination, which has limitations due to its ephemeral nature (Mukhopadhyay et al., 2018; Tizhoosh and Pantanowitz, 2018; Hanna et al., 2019).

Digitalization in pathology has been slower compared to other medical specialties, largely due to pathologists' reluctance to abandon traditional methods and the presence of barriers like regulatory and cost issues as well as "pathologist technophobia" (Hanna et al., 2019; Hanna and Pantanowitz 2019; Moxley-Wyles et al., 2020; Försch et al., 2021). Despite these challenges, AI has shown promise in enhancing diagnostic capabilities. Studies indicate that digital slide reviewing is as effective as manual methods (Loughrey et al., 2015; Elmore et al., 2017; Tabata et al., 2017). AI algorithms have been developed for detecting and classifying BC, achieving high

("Artificial intelligence" OR "Machine Learning" OR "Deep Learning" OR "Sentiment Analysis") AND ("breast cancer" OR "Breast Neoplasm" OR "Breast Tumor" OR "Mammary Neoplasm" OR "Mammary Carcinoma") AND (histo* OR cyto*).

**Figure 1.** Boolean search with keywords and their synonyms.

accuracy in differentiating between benign and malignant tumors (Cruz-Roza et al., 2017). A DL model differentiated benign and malignant tumors when tested on eight categories of images, four benign and four malignant, with an accuracy of 93.2% (Han et al., 2017). The breast cancer histology (BACH) challenge demonstrated that AI could achieve accuracy levels comparable to pathologists, improving overall diagnostic performance and interobserver concordance (Polonia et al., 2021). Additionally, DL models have successfully identified markers in BC and utilized nuclear characteristics to predict risk categories for patients (Romo-Bucheli et al., 2016; Lu et al., 2018; Whitney et al., 2018).

For example, the visual assessment of mitotic figures in BC histological sections stained with hematoxylin and eosin (H&E), referred to as the mitotic score, serves as the gold standard method for evaluating the proliferative activity of BC (Aleskandarany et al., 2012; van Dooijeweert et al., 2021) and pathologists face challenges in manually counting mitosis in histopathology slides, a process that is time-consuming (Cree et al., 2021). To address this, various contests such as the MITOSIS detection contest and others have facilitated advancements in automated counting methods (Aubreville et al., 2024). Recent studies have demonstrated that DL models can accurately count mitotic figures from H&E-stained slides of early stage ER-positive breast tumors, significantly reducing the time required for pathologists to read slides (Roux et al., 2013; ICPR, 2014; Romo-Bucheli et al., 2017; Veta et al., 2019). Notably, algorithms utilizing advanced architectures like ResNet-101, and Faster R-CNN have shown high accuracy, with one approach reducing reading time by 27.8% (Pantanowitz et al., 2020). Furthermore, a comprehensive automated system developed by Nateghi et al. (2021) can identify regions of interest for high mitotic activity, count mitoses from WSI, and predict tumor proliferation scores, outperforming previous methods. Although these AI models have yet to be deployed in formal clinical practice, the integration of digitalization and AI in pathology has the potential to enhance accuracy, reduce human errors and optimize the time needed for pathologists to review slides, ultimately benefiting both pathologists and patients (Aeffner et al., 2019; Kim et al., 2022).

This systematic review aims to examine AI models and their effectiveness in diagnosing BC, taking into account existing problems in the field of pathological diagnosis as well as the potential advantages of incorporating AI. Additionally, it investigates the potential of AI models to offer second opinions and their integration into the pathology workflow.

## Methodology

### *Preferred Reporting Items for Systematic Reviews and Meta-Analyses statement*

This review process follows the guidelines set out in the Preferred Reporting Items for Systematic Reviews and Meta-Analyses (PRISMA) statement, which was first developed in 2009 and updated in 2020. PRISMA functions as a framework specifically created to standardize the process of conducting systematic reviews and improve the thoroughness of their reporting (Page et al., 2021).

### *Search strategy*

A comprehensive literature search was carried out across three electronic databases: PubMed, EMBASE and Cochrane Library, to identify original articles that met the specified inclusion and exclusion criteria and were published up to April 2024. The search involved the use of keywords, their synonyms and Boolean operators, as illustrated in Figure 1. Additionally, the bibliographies of all relevant articles were meticulously reviewed to review additional studies that could potentially be included in the analysis. The titles were meticulously screened using the predetermined inclusion criteria, focusing on studies that assessed the application of AI in BC diagnosis. Notably, no restrictions were placed on publication year, country of origin or age of the studies. Subsequently, a full-text screening was carried out to include the most pertinent research papers for subsequent data extraction and analysis.

### *Selection process*

The literature search results were screened in a two-step process. Initially, the titles and abstracts of all articles were assessed for eligibility. After identifying relevant articles, full-text screening was conducted for the studies that met the eligibility criteria. The screening was done by applying the predefined inclusion criteria.

### *Eligibility criteria*

All research papers using AI in BC diagnosis or staging in comparison with pathologists' report or known datasets as a reference test were included. All types of original studies (either prospective or retrospective) containing their own data on AI validation or development and validation were included. The studies published in English language were included. We further excluded other types of publications such as reviews, single cases, editorial material, books, comments and papers in languages other than English, Articles dealing with other malignancies, articles that used AI to analyze data other than histological or cytological images (e.g., MRI, mammography) and articles with non-AI approaches for diagnosis (i.e., slide flow).

### *Inclusion criteria*

The study included

- BC diagnosis based on histopathology.
- AI models and their diagnostic accuracy, sensitivity, specificity and area under the curve (AUC) in BC diagnosis.
- Potential of AI models to be integrated in regular practice, provide second opinion in BC diagnosis.

### Exclusion criteria

- Following studies were excluded.
- Articles used AI to analyze data other than histological or cytological images (e.g., MRI, mammography).
- Articles with non-AI approaches for diagnosis (i.e., slide flow).
- Review papers, case studies, editorials, book chapters and commentary.
- Papers in languages other than English.
- Articles dealing with other malignancies.

Data from the included studies was extracted and recorded in a standardized data extraction sheet. The extracted data encompassed two main categories: (1) characteristics of the included studies and (2) outcome measures.

### Results

### Literature search and screening

A thorough search of the three main databases, namely PubMed, Cochrane CENTRAL and EMBASE, resulted in a total of 3113 records. After removing duplicates and irrelevant records, 1849 unique studies were assessed for eligibility based on their titles and abstracts. Over 1500 (1517) studies that did not meet our inclusion criteria were excluded. The abstracts of the remaining 332 articles were obtained for further evaluation. Application of predefined criteria led to the exclusion of 258 studies. The full text of 74 articles was reviewed out of which 47 were excluded due to the lack of relevant outcomes, inappropriate study design or poor quality. Finally, 27 studies were included in the systematic review. The flow diagram of the literature search and screening process is shown in Figure 2.

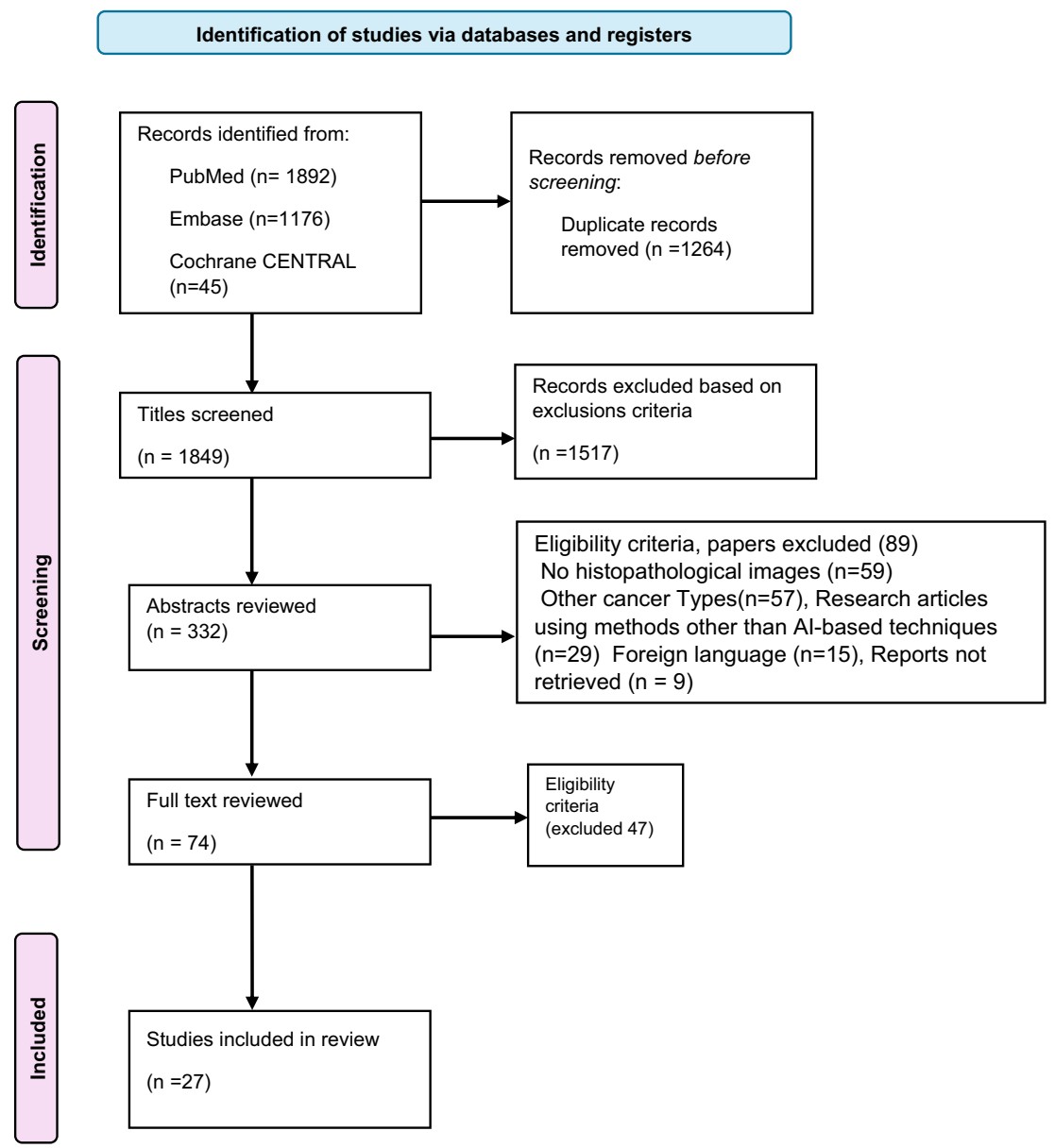

**Figure 2.** Flowchart describing the literature inclusion process.

### Characteristics of included studies

Twenty-seven research papers were analyzed in a thorough literature analysis. These studies specifically investigated the development of AI algorithms for diagnosing BC using histopathology images. The training data for histopathology images used in these research studies showed substantial variety, with data obtained from 21 to more than 400 patients. The literature evaluation includes papers published from 2017 to 2024, originating from various geographical regions including Korea, India, Egypt, the Netherlands, Brazil, Spain, Australia, Turkey, Poland, Japan, China and Malaysia. The research employed a range of AI techniques, with CNNs being particularly prominent. These CNNs were typically used in combination with other methods such as support vector machines (SVMs) or ensemble learning approaches. The datasets used for training and validation purposes included BreakHis (Spanhol et al., 2016), BACH (Aresta et al., 2019) and (*ICIAR, 2018 – Grand Challenge*), which are well-known. The pixel resolutions of the images varied in the investigations, which affected the level of detail and the potential effectiveness of the models. In addition, the performance metrics in the included studies showed variation, with accuracy, F1-score and AUC being the most reported. Table 1 provides a concise summary of the fundamental features of the research.

### Summary of findings

#### Breast lesion categorization

Researchers have developed innovative AI models for classifying breast lesions as benign or malignant, with the potential to identify histological subtypes (Singh et al., 2024). The SegEIR-Net model combines segmentation and classification techniques using EfficientNet, InceptionNet and ResNet, achieving high accuracies on the BreakHis dataset (up to 98.66%) and strong results on the BACH and UCSB datasets (93.33% and 96.55%, respectively) (Singh et al., 2024). Additionally, the Multilevel Context and Uncertainty aware (MCUa) model categorizes breast histology images into four types: normal tissue, benign lesion, in situ carcinoma and invasive carcinoma. The MCUa model demonstrated impressive performance, with static ensemble accuracy reaching 95.75% and dynamic accuracy reaching 98.11% on the BACH dataset, and outstanding results on the BreakHis dataset (up to 100% accuracy) (Senousy et al., 2022).

The context-aware stacked CNN (CAS-CNN) has demonstrated strong performance in classifying breast WSIs, achieving an AUC of 0.962 in distinguishing normal or benign slides from malignant ones (DCIS and IDC). The system exhibited a precision of 89.1% for categorizing WSIs and an overall accuracy of 81.3% in a three-class classification involving normal/benign, DCIS and IDC categories. While it effectively differentiated between normal/benign and IDC slides, it faced challenges in distinguishing between normal/benign and DCIS slides, as well as between DCIS and IDC slides (Bejnordi et al., 2017). Additional details on studies classifying breast lesions are provided in Tables 1–3.

#### Molecular subtyping

After the identification of a malignant breast lesion, immunohistochemistry is commonly performed to ascertain the molecular subtype. This procedure involves analyzing the levels of ER, PR, Her2 and the Ki67 mitotic index to determine the subtype, which can include luminal A, luminal B, Her2-enriched or triple negative, among other possibilities. The results of the studies regarding the use of AI models for molecular subtyping, with or without Ki67 measurement, have been summarized in Table 4. The AI models have exhibited an impressive accuracy rate (AUC of 0.75–0.91 vs. 0.67–0.8) when compared with conventional multiple instance learning models (Bae et al., 2023) and approximately 90% for an automated BC classification system utilizing SVM (Aswathy and Jagannath 2021).

#### Mitotic index assessment and quantification

Several studies have aimed to create models for identifying the mitotic proliferation index (Ki67) in BC. A notable approach is the FMDet method, designed to detect mitotic rates in breast histopathology images while addressing the domain shift problem caused by variability across different scanners (Wang et al., 2023). To enhance model applicability, two key strategies were implemented:

1. A novel data augmentation technique using Fourier analysis was introduced, which alters the frequency characteristics of training images to generate diverse samples that reflect real-world datasets. This involves replacing the low-frequency components of a source image with those from a reference image from a different domain.

2. Pixel-level annotations, specifically "instance masks," were utilized for identifying mitotic figures. These masks, derived from the Mitosis Domain Generalization challenge (MIDOG) 2021 bounding box data and a pretrained nucleus segmentation model (HoVer-Net), improved detection accuracy by allowing the network to capture subtle morphological differences in mitotic cells, surpassing traditional bounding box annotations (Wang et al., 2023). The model outperformed all other submissions in the challenge, with an F1 score of 0.77 (Wang et al., 2023).

A DL model specifically targeted for identifying mitotic proliferation in breast histopathology images has shown considerable potential (Saha et al., 2018. The precision, recall and *F*-score measures of this model were 92%, 88% and 90%, respectively. Significantly, the model's performance improved when hand-crafted features were integrated into its DL architecture. The model was trained and tested using datasets from MITOS-ATYPIA-14, ICPR-2012 and AMIDA-13. The study's findings highlight the model's high true positive rate, which indicates its precise ability to identify mitotic cells. Details of included studies focusing on training models for detecting and quantifying mitotic proliferation index have been tabulated in Table 5.

### Obstacles to widespread adoption of AI

Despite the remarkable outcomes demonstrated by the examined AI models in research contexts, huge challenges and constraints persist that hinder the extensive use of AI on a broader scale (Soliman et al., 2024). A primary procedural drawback of supervised AI models is their need of extensive, annotated datasets for training. Manual annotation is labor-intensive and exhibits variability both among and across pathologists, undermining the fundamental objective of AI models. Similarly, the intraclass heterogeneity and dependence on binary categorization during training, as previously mentioned, constitute additional barriers that may impact the efficacy of AI models. Furthermore, the lack of a defined area size for assessing Ki67 may lead to either an overestimation or underestimation of Ki67 activity. Moreover, certain preanalytic variables, including suboptimal sample quality, air bubbles, staining artifacts, unexpected staining patterns and discrepancies in interlaboratory sample preparation and staining

**Table 1.** Summary of studies categorizing breast lesions (i.e., benign vs. malignant)

| Author | Year | Country | AI model | Training and validation dataset | Model output | No. patients/ images used in training | Pixel level | Performance of the model | External validation dataset | Results of external validation |
|---|---|---|---|---|---|---|---|---|---|---|
| Amin et al. | 2023 | Korea | FabNet | BreakHis dataset | 1) Benign vs. malignant 2) Benign/ malignant lesion histological classification | 58 patients | 700 × 460 pixels | 99% Accuracy and a 98.9% F1 score for binary classification at 40× magnification | NR | NR |
| Bejnordi et al. | 2017 | Netherlands | Context-aware stacked CNN | Dataset of 157 WSIs (118 for training and 39 for validation) of breast tissue sections from the pathology archive of Radboud University Medical Center | 1) Benign vs. malignant 2) Normal/ benign vs. in situ vs. invasive samples | 118 WSIs | 0.243 μm × 0.243 μm pixel size | Model achieved an accuracy of 81.3% and a kappa value of 0.7 | Digitized WSIs selected for validation | Maximum accuracy of 0.9135 for the CAS-CNN 0.9241 for the WRN–4–2 |
| Gandomkar et al. | 2018 | Australia | ResNet–152 | BreakHis dataset | 1) Benign vs. malignant 2) Benign/ malignant lesion histological classification | 70 patients | 700 × 460 pixels | Achieved an average CCR of 98.10% for classifying images as benign or malignant and 95.15% for classifying images into eight classes (four benign and four malignant subtypes) | BreakHis | NR |
| Kolla et al. | 2024 | India | Modified pretrained Tiny Swin-Transformer V2 | BreakHis dataset | 1) Benign vs. malignant 2) Benign/ malignant lesion histological classification | NR | Resized to 256 × 256 pixels during preprocessing | Accuracy of 98.27%, 97.95%, 98.97% and 99.16% in the eight-groups, malignant, benign and binary, respectively | BreakHis | Best validation accuracy of 98.27% |
| Murthy and Balaji et al. | 2022 | NR | CNN | NR | 1) Benign vs. malignant 2) Benign/ malignant lesion histological classification | NR | 220 × 220 pixels | 1) For benign class ACC = 0.7866, TPR = 0.7921, TNR = 0.7837, FPR = 0.2163, PPV = 0.6597, NPV = 0.8769 2) For malignant class ACC = 0.7849, TPR = 0.788, TNR = 0.7832, FPR = 0.2168, PPV = 0.673, NPV = 0.8671 | NR | NR |
| Nahid et al. | 2018 | Brazil | CNN, LTSM, combined CNN and LTSM | BreakHis | Benign vs. malignant | 82 patients | 760 × 460 pixels | Accuracy of 77.4% for model 3, with a sensitivity of 95% and specificity of 59% – on using both the MS cluster algorithm and the SVM classifier together at 40× dataset | NR | NR |

(Continued)

**Table 1.** (*Continued*)

| Author | Year | Country | AI model | Training and validation dataset | Model output | No. patients/ images used in training | Pixel level | Performance of the model | External validation dataset | Results of external validation |
|---|---|---|---|---|---|---|---|---|---|---|
| Srikantamurthy et al. | 2023 | Korea | Hybrid CNN-LSTM ImageNet model | BreakHis dataset | 1) Benign vs. malignant 2) Benign/ malignant lesion histological classification | 82 patients | NR | Hybrid CNN-LSTM model achieved 99% accuracy for binary classification (benign vs. malignant) and 92.5% accuracy for multiclass classification (subtypes of benign and malignant) | NR | Accuracy of 99.03% at 40× images for binary classification, and 96.5% for multiclass classification |
| Stanitsas et al. | 2020 | NR | WAID built upon a CKD | FABCD | 1) Benign vs. malignant 2) Benign/ malignant lesion histological classification | 30 Patients | 10,000 × 9,000 pixels that are further divided into 150 × 150 pixel patches | Using LERM for similarity calculations, achieved the highest performance on the FABCD: 92.83% accuracy and 0.98 AUC | BreakHis | 92% CCM rate at the image level and 91.27% CCM at the patient level |
| Xu et al. | 2022 | China | MFSCNet, which is an improved DenseNet driven by an attention mechanism | BreakHis dataset | 1) Benign vs. malignant 2) Benign/ malignant lesion histological classification | 82 patients | 224 × 224 pixels | The best performing MFSCNet model (MFSCNet A) achieved 99.05%–99.89% accuracy in binary classification (benign or malignant). In multiclass classification (across 8 subtypes), the accuracy ranged from 94.36% to 98.41% | NR | NR |
| Zhu et al. | 2019 | China | CNNs | BreakHis, BACH | Benign (including normal/ benign lesions) vs. cancerous (in situ/ invasive lesions) | 20 WSIs in the BACH WSI dataset | 40,000 × 60,000 for the BACH WSI dataset | 87.5% patient-level and 84.4% image-level accuracy | BACH WSI dataset | BACH WSI dataset, the multimodel ensemble achieved 87.5% patient-level accuracy and 84.4% image-level accuracy |

AUC, area under the curve; BACH, breast cancer histology challenge; BreakHis, breast cancer histopathological database; CCM, correctly classified malignant; CCR, correct classification rate; CKD, covariance-kernel descriptor; CNNs, convolutional neural networks; DCNNs, deep CNNs; DenseNet, dense neural network; FABCD, fully annotated breast cancer database; LERM, log-Euclidean Riemannian metric; MFSCNet, classification of mammary cancer fusing spatial and channel features network; MLP, multilayer perceptron; QDA, quadratic discriminant analysis; ResNet-152, deep residual network with 152 layers; RF, random forest; SVM, support vector machine; WAID, weakly annotated image descriptor; WSIs, whole-slide images.

**Table 2.** Summary of studies categorizing breast lesions (i.e., normal/benign/in situ/invasive)

| Author | Year | Country | AI model | Training and validation dataset | Model output | No. patients/ images used in training | Pixel level | Performance of the model | External validation dataset | Results of external validation |
|---|---|---|---|---|---|---|---|---|---|---|
| Attallah et al. | 2021 | Egypt | Histo-CADx, a system that utilizes multiple AI models: Five CNNs: AlexNet, GoogleNet, ResNet–50, ShuffleNet and Inception-ResNet V2 Autoencoder (AE): For feature fusion and dimensionality reduction Classifiers: SVM, RF and QDA | BreakHis and ICIAR 2018 datasets | Normal vs. benign vs. in situ vs. invasive samples | 82 patients for BreakHis dataset and 400 WSIs for the ICIAR 2018 dataset | 2048 × 1536 pixels | Accuracy up to 99.54% | ICIAR 2018 dataset | Accuracy up to 97.93% |
| Bejnordi et al. | 2017 | Netherlands | Context-aware stacked CNN | Dataset of 157 WSIs (118 for training and 39 for validation) of breast tissue sections from the pathology archive of Radboud University Medical Center | 1) Benign vs. malignant 2) Normal/benign vs. in situ vs. invasive samples | 118 WSIs | 0.243 μm × 0.243 μm pixel size | Model achieved an accuracy of 81.3% and a kappa value of 0.7 | Digitized WSIs selected for validation | Maximum accuracy of 0.9135 for the CAS-CNN 0.9241 for the WRN–4–2 |
| Carvalho et al. | 2020 | Brazil | Several machine learning models: SVM RF, MLP, Xgboost | BACH 2018 dataset | Normal vs. benign vs. in situ vs. invasive samples (with different subgroup analyses) | 400 WSIs | 2040 × 1536 pixels | SVM performed best, with accuracy of 97.5% in differentiating normal vs. abnormal samples, and 100% in differentiating between benign vs. malignant samples | BACH 2018 dataset | Best performing model was SVM with PDIs achieving 95% accuracy and 96% precision |
| Fondón et al. | 2018 | Spain | SVM classifier with a quadratic kernel | High-resolution images from anonymous breast tumor biopsies | Normal vs. Benign vs. In Situ vs. Invasive Samples | 120 WSIs | 2048 × 1536 pixels | 75.8% accuracy using fivefold cross-validation on the training dataset | 20 images of similar complexity to the training dataset, and 16 images considered "extremely difficult" due to artifacts and | 1) Dataset with 20 images: 75% accuracy 2) Dataset with the 16 extremely difficult images: 61.11% accuracy |

(*Continued*)

**Table 2.** (Continued)

| Author | Year | Country | AI model | Training and validation dataset | Model output | No. patients/ images used in training | Pixel level | Performance of the model | External validation dataset | Results of external validation |
|---|---|---|---|---|---|---|---|---|---|---|
| | | | | | | | | | ambiguous malignancy | |
| Gecer et al. | 2018 | Turkey | FCNs followed by CNN | 240 digital breast histopathology images collected as part of NIH-sponsored projects | Benign vs. in situ vs. invasive samples | 180 WSIs | 535 × 416 pixels | Accuracy of 55% | External validation dataset consisted of independent diagnoses from 45 pathologists who evaluated the same 60 test slides used to assess the model's performance | 55% accuracy was comparable to the average accuracy of the 45 pathologists |
| Kanavati et al. | 2022 | Japan | EfficientNetB1 and EfficientNetB3 | Biopsies collected from three different hospitals and assessed by surgical pathologist | Benign vs. in situ vs. invasive samples | 3304 WSIs | Tile size of 224 × 224 pixels and a stride of 112 × 122 pixels | AUCs ranging from 0.95 to 0.99.8 | Biopsies diagnosed with pathologist from hospitals | Hospital 1 (Biopsy): AUC = 0.980, Log loss = 0.269 Hospital 1 (Surgical): AUC = 0.958, Log loss = 0.377, Hospital 2 (Surgical): AUC = 0.994, Log loss = 0.180, TCGA (Surgical): AUC = 1.000, Log loss = 0.274 |
| Laxmisagar et al. | 2022 | India | MobileNet AI architecture model | ICIAR BACH challenge dataset | Normal vs. benign vs. in situ vs. invasive samples | NR | 2040 × 1536 pixels | 87.5% accurate | NR | NR |
| Liu et al. | 2022 | China | MVMS-PFENet | BACH | Normal vs. benign vs. in situ vs. invasive samples | 400 for training, and 100 for testing | 2048 × 1536 pixels | f1 higher than 92.8% | Hyprid deep neural network | f1 score higher than 92% and accuracy reached 95% |
| Murtaza et al. | 2021 | Malaysia | EBrT-Net, which is based on AlexNet, but with modifications | BCBH dataset | Normal vs. benign vs. in situ vs. invasive samples | 249 WSIs | 258 × 258 pixels | Ranging from 87.50% to 100% for the four subtypes of BrT | BreakHis dataset | For the NC$_2$Net model with the best performance: Least validation loss achieved was 0.1281, corresponding to a |

(*Continued*)

**Table 2.** (*Continued*)

| Author | Year | Country | AI model | Training and validation dataset | Model output | No. patients/ images used in training | Pixel level | Performance of the model | External validation dataset | Results of external validation |
|---|---|---|---|---|---|---|---|---|---|---|
| | | | | | | | | | | validation accuracy of 95.66% |
| Senousy et al. | 2022 | NR | MCUa dynamic deep learning ensemble model | BACH dataset | Normal vs. benign vs. in situ vs. invasive samples | 39 patients | 2048 × 1536 pixels | 98.11% accuracy using the dynamic ensemble method | BreakHis | 99.80% accuracy using the static ensemble technique. The model achieved 100%, 99.95% and 99.90% accuracy using dynamic ensemble on δ values of 0.001, 0.003 and 0.03 |
| Singh et al. | 2024 | India | SegEIR-Net | BreakHis, BACH and UCSB | Breast cancer histopathological classification | NR | NR | 1) BACH dataset: 93.33%, UCSB dataset: 96.55% 2) BreakHis dataset: • 40X Magnification: 98.66%, • 100X Magnification: 98.39%, • 200X Magnification: 97.52%, • 400X Magnification: 95.22%" | NR | NR |
| Yang et al. | 2019 | China | Used an ensemble of three pretrained DCNNs: DenseNet–161, ResNet–152 and ResNet–101. This ensemble model is referred to as EMS-Net | * Training: BACH dataset * Validation: For validation, 40 images were randomly selected from the training dataset | Normal vs. benign vs. in situ vs. invasive samples | 400 WSIs | 2048 × 1536 pixels | EMS-Net algorithm achieved an overall accuracy of 91.75 ± 2.32% in a fivefold cross-validation using the training images from the BACH dataset | 100 microscopy images from the BACH dataset | EMS-Net model achieved an accuracy of 90% |
| Zhu et al. | 2019 | China | CNNs | BreakHis, BACH | Benign (including normal/benign lesions) vs. cancerous (in situ/invasive lesions) | 20 WSIs in the BACH WSI dataset | 40,000 × 60,000 for the BACH WSI dataset | 87.5% patient-level and 84.4% image-level accuracy | BACH WSI dataset | BACH WSI dataset, the multimodel ensemble achieved 87.5% patient-level accuracy and 84.4% image-level accuracy |

AUC, area under the curve; BACH, breast cancer histology challenge; BCBH, bioimaging challenge 2015 breast histology; BreakHis, breast cancer histopathological database; CCM, correctly classified malignant; CCR, correct classification rate; CKD, covariance-kernel descriptor; CNNs, convolutional neural networks; DCNNs, deep CNNs; DenseNet, dense neural network; FABCD, fully annotated breast cancer database; FCNs, fully convolutional networks; LERM, log-Euclidean Riemannian Metric; MCUa, multilevel context and uncertainty aware; MFSCNet, classification of mammary cancer fusing spatial and channel features network; MLP, multilayer perceptron; PDIs, phylogenetic diversity indices; QDA, quadratic discriminant analysis; ResNet-152, deep residual network with 152 layers; RF, random forest; SimCLR, simple framework for contrastive learning of visual representations; SOA, state-of-the-art; SSL, self-supervised learning; SVM, support vector machine; UCSB, University of California Santa Barbara; WAID, weakly annotated image descriptor; WSIs, whole-slide images; Xgboost, eXtreme Gradient Boost.

**Table 3.** Summary of studies assessing different histopathological subtypes of both benign and malignant breast lesions

| Author | Year | Country | AI model | Training and validation dataset | Model output | No. patients/ images used in training | Pixel level | Performance of the model | External validation dataset | Results of external validation |
|---|---|---|---|---|---|---|---|---|---|---|
| Amin et al. | 2023 | Korea | FabNet | BreakHis dataset | 1) Benign vs. malignant 2) Benign/ malignant lesion histopathological classification | 58 patients | 700 × 460 pixels | 99% accuracy and a 98.9% F1 score for binary classification at 40× magnification | NR | NR |
| Gandomkar et al. | 2018 | Australia | ResNet–152 | BreakHis dataset | 1) Benign vs. malignant samples 2) Benign/ malignant lesion histopathological classification | 70 patients | 700 × 460 pixels | Achieved an average CCR of 98.10% for classifying images as benign or malignant and 95.15% for classifying images into eight classes (four benign and four malignant subtypes) | BreakHis | NR |
| Kolla et al. | 2024 | India | Modified pretrained tiny swin-transformer V2 | BreakHis dataset | 1) Benign vs. malignant samples 2) Benign/ malignant lesion histopathological classification | NR | Resized to 256 × 256 pixels during preprocessing | Accuracy of 98.27%, 97.95%, 98.97% and 99.16% in the eight-groups, malignant, benign and binary, respectively | BreakHis | Best validation accuracy of 98.27% |
| Murthy and Balaji | 2022 | NR | CNN | NR | 1) Benign vs. malignant samples 2) Benign/ malignant lesion histopathological classification | NR | 220 × 220 pixels | 1) For benign class ACC = 0.7866, TPR = 0.7921, TNR = 0.7837, FPR = 0.2163, PPV = 0.6597, NPV = 0.8769 2) For malignant class ACC = 0.7849, TPR = 0.788, TNR = 0.7832, FPR = 0.2168, PPV = 0.673, NPV = 0.8671 | NR | NR |
| Singh et al. | 2024 | India | SegEIR-Net | BreakHis, BACH and UCSB | Breast cancer histopathological classification | NR | NR | 1) BACH dataset: 93.33%, UCSB dataset: 96.55% 2) BreakHis dataset: • 40X Magnification: 98.66%, • 100X Magnification: 98.39%, • 200X Magnification: 97.52%, 400× magnification: 95.22%" | NR | NR |
| Srikantamurthy et al. | 2023 | Korea | Hybrid CNN-LSTM ImageNet model | BreakHis dataset | 1) Benign vs. malignant samples 2) Benign/ malignant lesion histopathological classification | 82 patients | NR | Hybrid CNN-LSTM model achieved 99% accuracy for binary classification (benign vs. malignant) and 92.5% accuracy for multiclass classification (subtypes of benign and malignant) | NR | Accuracy of 99.03% at 40× images for binary classification, and 96.5% for multiclass classification |

**Table 3.** (*Continued*)

| Author | Year | Country | AI model | Training and validation dataset | Model output | No. patients/ images used in training | Pixel level | Performance of the model | External validation dataset | Results of external validation |
|---|---|---|---|---|---|---|---|---|---|---|
| Stanitsas et al. | 2020 | NR | WAID built upon a CKD | FABCD | 1) Benign vs. malignant samples 2) Benign/ malignant lesion histopathological classification | 30 patients | 10,000 × 9000 pixels that are further divided into 150 × 150-pixel patches | Using LERM for similarity calculations, achieved the highest performance on the FABCD: 92.83% accuracy and 0.98 AUC | BreakHis | 92% CCM rate at the image level and 91.27% CCM at the patient level |
| Umer et al. | 2022 | NR | 6B-Net a novel deep CNN mode, Additionally, they use a pretrained RESNET–50 model for feature extraction | BreakHis dataset | Benign/malignant lesion histopathological classification | 82 patients in the BreakHis dataset and 3771 WSIs the breast cancer pathology dataset | 2048 × 1536 pixels | 1) BreakHis dataset achieves a multiclass average accuracy of 90.10% with a training time of 147 s using the ESKNN classifier. 2) Breast cancer pathology dataset method achieves a multiclass average accuracy of 94.20% with a training time of 226 s using the ESD classifier" | Breast cancer pathology dataset | A maximum 93.60% accuracy using the ESD classifier and a minimum 83.00% accuracy using the EBT classifier |
| Xu et al. | 2022 | China | MFSCNet, which is an improved DenseNet driven by an attention mechanism | BreakHis dataset | 1) Benign vs. malignant samples 2) Benign/ malignant lesion histopathological classification | 82 patients | 224 × 224 pixels | The best performing MFSCNet model (MFSCNet A) achieved 99.05–99.89% accuracy in binary classification (benign or malignant). In multiclass classification (across 8 subtypes), the accuracy ranged from 94.36% to 98.41% | NR | NR |

AUC, area under the curve; BACH, breast cancer histology challenge; BCBH, bioimaging challenge 2015 breast histology; BreakHis, breast cancer histopathological database; CCM, correctly classified malignant; CCR, correct classification rate; CKD, covariance-kernel descriptor; CNNs, convolutional neural networks; DCNNs, deep CNNs; DenseNet, dense neural network; FABCD, fully annotated breast cancer database; FCNs, fully convolutional networks; LERM, log-Euclidean Riemannian Metric; MCUa, multilevel context and uncertainty aware; MFSCNet, classification of mammary cancer fusing spatial and channel features network; MLP, multilayer perceptron; PDIs, phylogenetic diversity indices; QDA, quadratic discriminant analysis; ResNet-152, deep residual network with 152 layers; RF, random forest; SimCLR, simple framework for contrastive learning of visual representations; SOA, state-of-the-art; SSL, self-supervised learning; SVM, support vector machine; UCSB, University of California Santa Barbara; WAID, weakly annotated image descriptor; WSIs, whole-slide images; Xgboost, eXtreme Gradient Boost.

**Table 4.** Summary of studies assessing breast cancer molecular subtyping (i.e., according to estrogen receptors (ER), progesterone receptors (PR) and Her2 – with or without ki67 mitotic index analysis)

| Author | Year | Country | AI model | Training and validation dataset | Model output | No. patients/ images used in training | Pixel level | Performance of the model | External validation dataset | Results of external validation |
|---|---|---|---|---|---|---|---|---|---|---|
| Aswathy et al. | 2021 | India | SVM | UCSB dataset and validated by BreakHis dataset | Breast cancer molecular classification (ER, PR, Her2) | 58 WSIs | 896 × 768 pixels | Accuracy: 91%, Sensitivity: 92%, Specificity: 90.3%, F-score: 89.9%, Precision: 88.9%, Balanced Accuracy: 91.2% | BreakHis dataset | SVM model achieved an average accuracy of 89.1% on the BreakHis dataset |
| Bae et al. | 2023 | Korea | 3DHistoNet, which utilizes SSL approach called SimCLR | Patients with primary breast carcinoma diagnosed between 2018 and 2020 at the Department of Pathology, National Cancer Center, South Korea | Breast cancer molecular classification (including Ki67 mitotic index) | 401 patients | 0.25 μm | AUC ranging from 0.75 to 0.91 | NR | NR |

AUC, area under the curve; BCBH, bioimaging challenge 2015 breast histology; BreakHis, breast cancer histopathological database; MLP, multilayer perceptron; SVM, support vector machine; UCSB, University of California Santa Barbara; WSIs, whole-slide images; Xgboost, eXtreme.

techniques, remain unavoidable to this day. Consequently, each AI tool necessitates validation and verification under these specific conditions. The identified variables may influence the accuracy of specimen analysis conducted by the AI model, resulting in incorrect outcomes. Additionally, significant economic and regulatory challenges persist regarding the implementation of AI technology in pathology laboratories on a global or national scale (van Diest et al., 2024). The concern regarding AI's potential to replace pathologists raises ethical issues, as noted by Moxley-Wyles et al. (2020) and van Diest et al. (2024). Critics, comprising both individuals and governmental bodies, articulate concerns regarding the notion that a patient's treatment may be dictated by an AI analysis. This is particularly concerning as most existing algorithms remain in the early stages of development, having been tested on limited populations and lacking adequate safety data. The broad adoption of this technology faces obstacles stemming from the diverse staining techniques employed across laboratories, the necessity for full digitalization prior to implementation, and the partial integration of Picture Archiving and Communication Systems with AI algorithms (van Diest et al., 2024). Addressing these concerns is crucial prior to the deployment of AI models in clinical practice.

### Addressing limited sample size

A study aimed at improving BC histopathology image classification addressed the challenge of limited slide image datasets through data augmentation and transfer learning (Zhu et al., 2019). The researchers developed a hybrid CNN that combines local and global visual inputs to capture detailed features and structure. They introduced a Squeeze-Excitation-Pruning block to reduce the model size without sacrificing accuracy. To enhance generalization, they used a multimodel assembly technique, training various models on different data subsets and merging their predictions. This approach

proved more effective than a single-model systems and outperformed existing methods on the BACH dataset (87.5% patient-level and 84.4% image-level accuracy), indicating its potential for real-world clinical use (Zhu et al., 2019).

Recent studies have explored the development of advanced models for classifying BC subtypes using histopathology images. A hybrid CNN-LSTM model achieved high accuracy in binary and multiclass classifications, with a binary accuracy ranging from 98.07% to 99.75% and multiclass accuracy from 88.04% to 96.5% (Srikantamurthy et al., 2023). Another study utilized a pretrained DenseNet-169 model on the BreakHis dataset, achieving 98.73% accuracy on the validation set and 94.55% on the test set, highlighting the importance of compatible-domain transfer learning – a method through which histological images are used in the pretraining of the model and then fine-tuned on a finite cytological target dataset (Shamshiri et al., 2023). Additionally, various studies emphasize the importance of feature selection and fusion to improve classification accuracy, noting that integrating DL features with handcrafted attributes can lead to better outcomes. Multiscale analysis, incorporating different image patch scales, also contributes to enhanced accuracy in classification tasks (Attallah et al., 2021; Liu et al., 2022).

### Training datasets

Another important consideration in the included studies is the choice of dataset and the use of transfer learning. Multiple studies have utilized the BreakHis dataset, which contains images of varying magnification levels, to advance BC diagnosis, despite challenges like small sample sizes and data variability (Gandomkar et al., 2018; Nahid et al., 2018; Carvalho et al., 2020). To address these issues, researchers have applied transfer learning techniques, initially training models on larger datasets like ImageNet before

**Table 5.** Summary of studies assessing the Ki67 mitotic index

| Author | Year | Country | AI model | Training and validation dataset | Model output | No. patients/ images used in training | Pixel level | Performance of the model | External validation dataset | Results of external validation |
|---|---|---|---|---|---|---|---|---|---|---|
| Bae et al. | 2023 | Korea | 3DHistoNet, which utilizes SSL approach called SimCLR | Patients with primary breast carcinoma diagnosed between 2018 and 2020 at the Department of Pathology, National Cancer Center, South Korea | Breast cancer molecular classification (including Ki67 mitotic index) | 401 patients | 0.25 μm | AUC ranging from 0.75 to 0.91 | NR | NR |
| Saha et al. | 2018 | NR | NR | MITOS-ATYPIA–14, ICPR–2012 and AMIDA–13 | 1) Mitotic detection 2) Mitotic figure counting | 11,921 training WSIs (256 batch size) | 71 × 71 pixels | Average precision of 0.92, a recall of 0.88, and an F-score of 0.90 | NR | 0.92% precision, 0.87 recall and 0.89 at fivefold cross-validation |
| Wang et al. | 2023 | China | FMDet, a novel mitosis detection algorithm based on a semantic segmentation approach. The core architecture is a U-Net framework | Training: MIDOG 2021 challenge training set, comprising 200 WSIs from four different scanners. Validation: The paper used an out-of-domain validation set extracted from the MIDOG21 training set | Mitotic figure counting | 200 WSIs | 8000 × 8000 pixels | F1 score = 0.7773 | Four external datasets were used: AMIDA13, MITOSIS14, TUPAC-auxiliary, MIDOG22 | AMIDA13: F1 score = 0.6791, Recall = 0.7405, Precision = 0.6271. TUPAC-auxiliary (Centers 2 and 3): F1 score = 0.7458, Recall = 0.8018, Precision = 0.6971. TUPAC-auxiliary (Centers 1, 2 and 3): F1 score = 0.6946, Recall = 0.7738, Precision = 0.6301. MITOSIS14: F1 score = 0.4901, Recall = 0.5564, Precision = 0.4380 MIDOG22: F1 score = 0.7389, Recall = 0.7796, Precision = 0.7022 |

AUC, area under the curve; SimCLR, simple framework for contrastive learning of visual representations; SSL, self-supervised learning; WSIs, whole-slide images.

fine-tuning them on BreakHis (Kanavati et al. 2022; Laxmisagar and Hanumantharaju, 2022; Liu et al., 2022; Xu et al., 2022). Additionally, compatible-domain transfer learning has been shown to boost model performance (Shamshiri et al., 2023). Various approaches have been explored to make up for insufficient training datasets, including the combination of multiple classifiers and a self-trained AI algorithm utilizing a three-stage analysis technique with a WSI stacking system (Bae et al., 2023). Overall, these efforts highlight the potential of DL and image analysis in enhancing BC diagnosis and prognosis.

## Discussion

Commercially available AI models specifically designed for BC detection exist (Soliman et al., 2024). The mentioned entities comprise Mindpeak, Owkin, Visiopharm, Paige Breast Suite and IBEX Galen Breast. Mindpeak is located in Hamburg, Germany (Abele et al., 2023); Owkin is based in Paris, France; Visiopharm is situated in Hovedstaden, Denmark (Shafi et al., 2022); Paige Breast Suite is established in New York, United States and IBEX Galen Breast is

positioned in Tel Aviv, Israel. These algorithms improve pathologists' consistency, precision and sensitivity while decreasing time demands (Soliman et al., 2024). Nonetheless, various limitations persist in obstructing its extensive implementation on a larger scale. AI-driven specimen analysis now demonstrates specific procedural drawbacks, as observed by Soliman et al. (2024). They also evaluated the contribution of AI in enhancing histological analysis of BC and to collectively assess the efficacy of each AI model, while also emphasizing potential limitations and downsides associated with each model.

## Tissue classification

Most studies in this review evaluated the efficacy of AI in accurately classifying breast tissue specimens, demonstrating significant precision in distinguishing normal (or benign) tissues from malignant ones (Amin et al., 2023; Attallah et al., 2021; Gandomkar et al., 2018; Singh et al., 2024). In the research conducted by Singh et al. (2024), the SegEIR-Net model achieved an accuracy surpassing 98% on the BreakHis dataset. Likewise, the Histo-CADx model

introduced by Attallah et al. (2021) has attained an accuracy of 99.54%. The initial study presenting the BreakHis dataset (Spanhol et al., 2016), aimed at evaluating AI model efficacy in tissue classification, attained an accuracy of approximately 97%. Han et al. (2017) conducted a study that attained an average accuracy of approximately 93% for eight-class classification (comprising four benign and four malignant categories) of the BreakHis dataset, in contrast to the traditional binary or ternary classification systems evaluated in most studies. The findings from these studies demonstrate that AI can continuously prove to be a dependable instrument for breast specimen classification.

Nonetheless, despite these encouraging results, many issues persist regarding tissue classification. For instance, there remains a notable heterogeneity in the datasets employed and the outcomes evaluated. Should research evaluate the efficacy of AI in distinguishing between normal and malignant? distinguishing benign from malignant? Perhaps, DCIS from invasive carcinoma (Amin et al., 2023)? In-class heterogeneity exists, and binary classification tasks for AI models (e.g., normal vs. abnormal) are frequently insufficient due to the presence of gray zones and unusual findings in clinical practice (Amin et al., 2023; Bejnordi et al., 2017). Furthermore, certain unusual subtypes exist but may not be included in training and/or testing datasets (Hatta et al., 2023). The presence of in-class heterogeneity and imbalanced datasets complicates the assessment of AI's accuracy in classifying each category, particularly the rare ones (Amin et al., 2023; Hatta et al., 2023). Similarly, Bejnordi et al. (2017) reported that although their model excelled in distinguishing benign from malignant tumors, it encountered difficulties with borderline categories such as DCIS. Additionally, Amin et al. (2023) also documented similar findings.

### *Molecular subtyping*

Current issues associated with manual molecular subtyping include frequent interpathologist discrepancies, particularly with Her2 status (Robbins et al., 2023). In contrast to specimen categorization, which relies mostly on visible microscopic features, subtyping is contingent upon supplementary technical factors, such as the extent of staining and/or manual enumeration in the computation of Ki67. Therefore, several studies are being conducted using different AI models to address the challenge of inconsistency (Abele et al., 2023).

Similar to the reported findings about AI's performance in tissue classification, the studies assessing molecular subtyping in this review have likewise yielded promising outcomes. The data from the studies reported in this review indicate that AI excelled in both molecular subtyping and Ki67 computation. The research conducted by Bae et al. (2023) attained a molecular subtyping accuracy of 91%. Aswathy et al. (2021) found a balanced accuracy of 91.2% for their SVM model in predicting ER, PR and Her2 status. Consistent with findings in this review, several other studies have documented significantly enhanced interpathologist agreement rates following the implementation of AI aid. AI enhanced interpathologist agreement rates from approximately 88% to 96% for the Ki67 score and from roughly 89% to 93% for the ER/PR status (Abele et al., 2023). Similarly, concerning Her2 status, a study by Jung indicated a significant rise in interpathologist agreement rates from approximately 49.3% to 74.1% (p < 0.001), with the use of AI driven ER/PR and Her2 analyzers Additionally, the concordance rates for ER and PR status improved, albeit to a lower degree (93.0–96.5% p = 0.096 for ER, 84.6–91.5%, p = 0.006 for PR) (Jung et al., 2024).

## Conclusion and future directions

Various AI tools may rapidly become an effective aid to histopathologists facing increasing demands for precise and speedy BC diagnosis. The identified drawbacks are paramount and need to be effectively addressed before we can reap the true benefits of this technology. It is recommended that future research endeavors focus on the following key areas to improve the performance and validity of existing AI models in the context of histopathological evaluation:

1. *Establishment of Standardized Datasets:* The creation of standardized, multi-institutional datasets that adhere to consistent preanalytic methodologies – such as sample preparation and tissue staining – should be prioritized to improve the generalizability of AI models across diverse clinical settings.
2. *Integration of Multimodal Data:* To enhance the predictive performance of AI systems, it is imperative to incorporate additional diagnostic modalities, including but not limited to imaging techniques and molecular profiling, into the analysis of histopathological specimens. This multimodal approach can offer a more comprehensive understanding of disease characteristics.
3. *Development of Ethical Protocols:* The formulation of robust ethical guidelines is essential to ensure the responsible application of AI technologies. This includes strategies for mitigating biases inherent in data and algorithms and enhancing transparency in the decision-making processes of AI systems.
4. *Improvement in Economic Viability:* It is crucial to explore cost-effective strategies for the implementation of AI solutions within clinical practice. An analysis of economic sustainability will ultimately support the broader adoption and integration of AI technologies in the healthcare sector.

By addressing these areas, future research can significantly contribute to the advancement of AI methodologies, ensuring they are both practical and ethically sound in clinical applications.

**Open peer review.** To view the open peer review materials for this article, please visit http://doi.org/10.1017/pcm.2025.10006.

**Data availability statement.** Data sharing not applicable – no new data generated.

**Author contribution.** AJ and AS made substantial contributions to the conception and design of the work. A.J. drafted the work and revised it critically for important intellectual content. A.S. was responsible for final revision and approval of the version to be published.

**Financial support.** This research received no specific grant from any funding agency, commercial or not-for-profit sectors.

**Competing interests.** The authors declare none.

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
