## [Reviewer Report]

Overall Impression:

I read this paper with great interest, as there is a lot of work being done aiming to improve histopathologists’ and cytopathologists’ performance when diagnosing cancer. Unfortunately, the way this paper is written gives the strong impression that the authors want to push the message that AI-based models are here to save breast cancer diagnosis from pathologists’ errors and inter-observer variability, even though most of the work they cite only talks about the “potential” of these models to help pathologists in the clinical practice. Now, the keyword here is “potential”, as for the most part they have not been implemented in pathology laboratories yet, and massive challenges still prevent wide adoption, such as the different staining processes used by different labs, the lack of representation of rare diseases in the datasets used to train AI-based models, the reduced representation of minority populations in the same databases, integration of the AI-models with the LIS, etc. None of that seems to reduce the authors’ enthusiasm about the use of AI-models in histopathology. While I admire their optimism, the way that they present their views for the most part suggests great naivete about how things work in the real world, but in a few instances it seems to misrepresent what is actually being said in the papers that they cite so that it conforms better to the authors’ message. I think that this is very unfortunate, and I would highly recommend the authors to (1) tone down their enthusiasm and stick with the facts; (2) don’t confuse “potential” with actual use of AI-based models; and (3) don’t dismiss the immense challenges that still prevent the wide adoption of these models in clinical labs.

Abstract:

Pp 1, lines 18-29. There is no need to go into such great details about breast cancer incidence, mortality rates and projected number of new cases by 2040 as this is not a review of the disease itself, it’s a review of the use of AI in the diagnosis of the disease. In my opinion, the entire Abstract needs to be rewritten and focused on what the paper is actually discussing.

Pp 1, lines 29-32. Please clarify what you mean by “traditional methods”. Are these pathologists? If so, please state that.

Pp 1, lines 35-37. I believe that it is too early to state that AI has “improved accuracy, efficiency, and consistency”, as it is not currently in use in the clinical practice, so it hasn’t really been tested with real data, with its high degree of variability, including different staining methods used by different labs, presence of artifacts in the images, possible presence of rare disease in the slide, etc. I understand that the authors are enthusiastic about the topic, but given the current situation of AI in histopathology, I do believe that there is cause for some moderation in one’s claims.

Introduction:

Pp 2, lines 17-20. The authors state that “AI refers to the utilization of technology and computers to imitate human-like cognitive processes and intelligent actions”. Well, AI is implemented by machine learning (ML) algorithms, and many of those have no resemblance whatsoever to “human cognitive processes”. As such, please rewrite this statement.

Pp 2, line 49. It should read “AI applications in medicine have evolved…”.

Pp 2, line 54. What are “traditional algorithm-based methods”??? Please clarify.

Pp 3, lines 3-5. As the authors were previously talking about AI, I am assuming that the sentence that lists the benefits of “predictive models” is also referring to those that use AI. However, in general, predictive models do not have to be based in deep learning (DL) algorithms, they can be based on traditional machine learning (ML) algorithms, such as Support Vector Machines, and some of those traditional algorithms perform quite well. Please clarify what the authors are referring to in this sentence.

Pp 3, line 14. Not all machine learning algorithms “make predictions”. Many of them are used in segmentation or classification tasks. Please reword.

Pp 3, line 19. What is “The term” referring to??? Please clarify.

Pp 3, line 26. It should read “…. predefined results; similarly….”

Pp 3, line 41. Not all neural networks “mimic brain function”. Most of them are inspired by biological neurological systems, but the actual resemblance is small.

Pp 3, line 44. The authors first need to define what they mean by “neuron”. Secondly, I do not understand when the authors say that “each neuron processes inputs”. Do they mean to suggest that all neurons are connected to the input layer? Or that they process inputs from the previous layer? Please clarify.

Pp 3, line 46. The authors state that “ANNs have one hidden layer”, but this is not true for all types of ANNs. For example, Adaptive Resonance Theory (ART) ANNs have no hidden layers. Please reword.

Pp 3, lines 53-55. The authors state that “Recently, there has been a rise in using these models for accurate diagnoses”. This is factually not correct. While there may have been a rise in the development of Deep Learning (DL) algorithms for disease diagnoses, for the most part they are not used in the clinical practice, which is what is suggested by what the authors wrote. Please rewrite the sentence to reflect the true state of current application of DL in the clinical practice.

Pp 4, line 6. CNNs are also design for “classification” tasks. Please include those.

Pp 4, line 17. Please provide a reference for use of AI in “genomic analysis” and in “patient monitoring”.

Pp 4, line 20. Please provide a reference for use of AI in “wearable health technology” and in enhancing “doctor-patient interactions”.

Pp 4, line 22. Please provide a reference for use of AI in enabling “remote therapy”.

Pp 4, line 25. The paper by Alowais et al, 2023 discuss the “potential” role of AI in health care, but they also report at length on the challenges that need to be addressed before AI can be implemented in the clinical practice. I do not see how, from that paper, the authors can conclude that “Integrating AI into healthcare can significantly improve the effectiveness, accuracy, and personalization of medical diagnosis”. Please clarify.

Pp 4, line 35. How can the authors state that AI “is significantly advancing cancer diagnosis” and cite as support a paper (Sufyan et al 2023) that only talks about the “potential” of AI to advance cancer diagnosis??? There is a huge difference between potential and actual implementation, and the authors seem to not recognize that. Please modify your statement.

Pp 4, line 35. The paper by Alshuhri et al 2024 is not included in the References.

Pp 4, lines 40-42. The authors seem enthusiastic about how AI is going to “transform patient care”, however they seem to forget that this is not the first time that machine learning has been used to aid physicians in their practices. In the late 1990s and in the 2000s, Computer-Aided Detection (CADe) was also a big promise, but it completely failed to deliver when incorporated in the clinical practice. In light of that, I would strongly recommend that the authors moderate their enthusiasm for AI until we have actual evidence that it is improving patient care.

Pp 4, lines 49-51. The authors state that CADe and CADx “play vital roles in medical imaging”. First of all, that is not true. The clinical implementation of CADe and CADx have been marked by undelivered promises and a significant number of False Positives per case, which erodes radiologists’ trust in the system and leads less-experienced observers, like radiology residents, astray. Please do a more thorough search of the literature, instead of just presenting one paper (He Z, 2020, which by the way is not listed in the References) that probably supports your view that CADe and CADx are actually wonderful.

Pp 4, line 56. Please provide a reference for the statement that AI is improving accuracy in “identifying cancer progression”. Also, provide a reference for the statement that AI is “aiding in early detection and diagnosis”.

Pp 5, line 35. The authors state that histopathological diagnosis is “still relying on microscopic evaluations by human pathologists”. This does not take into account the fact that a lot of Institutions have moved on to Digital Pathology, which should be stated here. Furthermore, false positive and false negatived errors are not going to be erased by digitalizing the process of assessing slides. Similarly, inter- and intra-pathologist variability is also not going to be erased by moving to Digital Pathology, as the authors seem to suggest in this paragraph.

Pp 6, lines 3-5. The process of cancer diagnosis starts with the preparation of the slides and it ends with the pathologists’ interpretation, determining if and which type of disease may be present. Please clarify in which part of this process AI “is essential for improving diagnostic processes”.

Pp 6, line 15. Please clarify what the authors mean by saying that computer monitors have “much greater clarity than traditional microscopy”.

Pp 6, line 17. Which unit is the “100,000 x 100,000” representing? Are they pixels? Please clarify.

Pp 6, line 20. The authors state that Digital Pathology reduces “interpretation errors”, but because in the previous page they only vaguely alluded to false positives and false negatives that resulted as pathologists used the traditional microscopes to interpret slides, without citing any actual error rates, it is difficult to visualize the magnitude of the effect that Whole Slide Imaging (WSI) would have on the reduction of interpretation errors. Could the authors please be more specific about that?

Pp 6, line 54. The authors say that in the BACH challenge “AI could achieve accuracy levels comparable to pathologists”. Can they please describe a bit more about who these “pathologists” were? Were they domain experts? Were they general pathologists? Please clarify.

Pp 7, line 3. Please provide a reference for the statement that AI improved “interobserver concordance”.

Pp 7, line 22. Please provide a reference for the MITOSIS detection contest.

Pp 7, line 40. In the system developed by Nateghi et al, which can “identify regions of interest”, what are these regions of interest for? Areas to count mitoses? Please clarify.

Pp 7, lines 45-47. I do not understand what the authors mean when they say that AI “optimizes the time needed for pathologists”. Please clarify.

Pp 8, line 6. Instead of “… pathology team”, it should read “… pathology workflow”.

Methodology:

Fine as written.

Results:

Pp 12, line 39. Please provide references for the BreakHis, BACH and ICIAR datasets.

Pp 12, lines 39-41. How are the authors assessing that these datasets are “significant”? Please clarify.

Pp 22, line 14. Instead of “study”, it should read “studies”.

Pp 22, line 19. The authors report that the AI models in Bae et al (2023) study exhibited “an impressive accuracy rate of approximately 91%”, but in reality Bae et al’s paper cited a range of AUC values ranging from 0.75-0.91. Hence, it is not correct for the authors to simply extract the highest number in that range and claim that this was the generalized performance.

Pp 22, line 30. Please include a reference for the issue of variability across different scanners.

Pp 22, lines 32-51. The authors cite two steps that were implemented to “enhance model applicability”. However, they fail to report what was the model’s performance at the end of the implementation of those 2 steps.

Pp 23, line 3. Were any statistical tests carried out to determine that the model’s performance was “significantly” improved when handcrafted features were integrated into the DL architecture? Please clarify.

Pp 23, line 8. Can the authors please show the change in the model’s performance from pre- to post-integration of handcrafted features in the DL architecture?

Pp 23, line 26. Please include a reference for the first statement in the paragraph.

Pp 23, lines 38-40. Can the authors please provide some additional information about the performance of the new approach they are discussing vs. that of single-model systems in the BACH dataset? It is difficult to take these statements at face value without seeing some numbers.

Pp 24, line 38. Please define what the authors mean by “compatible-domain transfer learning”.

Pp 24, lines 38-40. Please provide an actual numerical example that shows how model performance is improved by using this “compatible-domain transfer learning”.

Pp 24, lines 40-45. I do not understand what the authors are referring to in the sentence where they are describing that “Various approaches have been explored…”. These approaches have been explored to do what? Please clarify.

Discussion:

Pp 25, lines 26-28. The authors cite a number of commercially available AI-based software specifically designed for breast cancer diagnosis. They follow that by saying that “These algorithms improve pathologists’ consistency, precision, and sensitivity while decreasing time demands”, and cite as reference for this statement a paper by van Diest et al (2024). I read that paper, and nowhere in it the authors evaluate any commercially available AI-based algorithms for breast cancer diagnosis. Thus, the connection made by the authors between commercially available AI software and their results when used in the clinical practice is very misleading, as it is not supported by the reference cited by the authors.

Pp 25, lines 38-40. Where in this paper did the authors “assess the efficacy of each AI model while also emphasizing potential limitations and downsides associated with each model”??? On the contrary, the authors presented the AI-based models as if they were perfect and were already working in the clinical practice, all while citing papers that only talked about the “potential” of AI one day being used in the clinical practice. Please clarify where in the paper those assessments were carried out, and also were the limitations of each model were described.

Pp 26, lines 5-8. I do not understand what the authors mean when they say “These findings correspond with literature research that has also documented exceptional performance”. What are they talking about? Can they please clarify? Also, include references for the “literature research” cited?

Pp 26, lines 20-24. Please clarify how do the authors know that the excluded findings from your analysis “demonstrate that AI has continuously proven to be a dependable instrument for breast specimen classification”.

Pp 27, line 14. Please include a reference after the “… with Her2 status”.

Pp 27, line 43. Please clarify what you mean by “AI pathologist help”.

Pp 27, line 54. The authors say that “the concordance rates for ER and PR status improved, albeit to a lower degree”. Please provide actual numbers to make this a bit more concrete for the readers.

Pp 28, lines 25-28. What do the authors mean when they say that AI models have a “dependence on binary categorization during training”? Please clarify.

Pp 29, line 3. Instead of “ought”, please use “may”.

Pp 29, lines 17-19. Instead of “… in different laboratories”, please use “… in the clinical practice”.

Conclusions and Future Directions:

Pp 29, line 31. The authors state that AI tools are “rapidly becoming an effective aid to histopathologists”. However, how can this be true, considering that most AI-models are not currently used in the clinical practice for the variety of reasons explained in the papers cited by the authors (of which some were cited in the paragraph above this)? Please clarify.

Pp 29, lines 33-36. The authors state that “The identified drawbacks can be effectively addressed”. Do they have any understanding of the magnitude of the challenges to address the difficulties in implementing AI-based models in WSI? For example, (1) How does one standardize the staining protocols used by the different labs around the world? (2) How does one increase the number of samples of rare diseases, in order to train the models appropriately? (3) How does one increase representation from minority populations in the training sets used to train the AI models? And so forth. These are not trivial challenges. Please don’t treat them as if they are!

Pp 29, line 38. Instead of “augment”, please use “improve”.

Pp 29, line 44. The “Establishment of Standardized Datasets” is not that useful if the datasets are not freely available to all researchers to train, validate and test their models. Please include “freely available” as a condition sine-qua-non on the “Establishment of Standardized Datasets”.

Pp 30, lines 6-8. When the authors say that AI models should incorporate “additional diagnostic modalities”, and then they cite as an example “imaging techniques”, what do they mean by that? Do they mean the patients scans, like computed tomography or magnetic resonance imaging? Please clarify.

Pp 30, lines 16-18. What are the biases that are “inherent in data and algorithms”??? Algorithms by themselves will only become biased depending on the distribution of the data that is used to train them, so they don’t have any “inherent biases”. Please clarify.

Pp 30, line 24. Instead of “Improvement of Economic…”, it should read “Improvement in Economic…”

Pp 30, lines 31-33. This last sentence should not be attached to point 4, it should be presented by itself below point 4, as it relates to all 4 points presented above it.

---

## [Reviewer Report]

The application of AI in the diagnosis of breast cancer is a highly relevant and widely discussed topic in medical clinics and medical research.

The authors have compiled a comprehensive systematic literature review on this subject, providing a wealth of information. However, in its current form, the manuscript is overly lengthy and includes several basic concepts that are already well-established in the field. I recommend that the authors significantly condense the manuscript by streamlining the content—particularly in the Introduction and Discussion sections—by removing foundational information that does not add novel value.

Additionally, the Results and Findings sections should be summarized and made more concise to improve readability and focus. The manuscript would also benefit from a thorough review for grammatical errors, inconsistencies in sentence structure, and incorrect or undefined abbreviations. A more structured and succinct presentation will greatly enhance the impact and clarity of the paper.

---

## [Editor Report]

The manuscript is timely and relevant. However, as noted by reviewers there is some degree of over optimism, and lack of distinction between potential and actual deployment of AI pathology into clinical practice.

We would welcome review of the feedback provided and revisions to enhance the paper.

---

## [Reviewer Report]

Overall Impression:

I would like to commend the authors for taking into account so many of the comments/suggestions made in the previous review. I believe that the paper is significantly clearer now, and many of the grandiose (or over-optimistic) statements have been removed – although a few have stayed, as I point out in my review below. I think that the authors have done a good job presenting the different AI systems that they discuss, but there were some major points that left me a bit confused. First, the authors say that “3113 studies were potentially found, and after applying the inclusion criteria and filtering out the duplicates, 1194 unique studies were selected. From these, 1516 studies were excluded” due to a variety of factors. How can 1516 studies be excluded from 1194 unique studies? Second, and in my view a very important point, there is no presentation in the Results section of the Possible Obstacles for Implementation of AI in the Clinical Practice that may have been discussed in the studies reviewed. Instead, we get a sub-section in the Discussion that is titled “Other obstacles to widespread adoption of AI”. This is confusing, as there has not been any discussion on obstacles to widespread adoption of AI thus far. More detailed comments about each section follow below.

Abstract:

Pp 2, lines 19-22. It should read “… particularly those that have been difficult to discern through routine microscopy”.

Impact Statement:

Pp 3, line 22. Instead of “breast cancer medication”, it should read “breast cancer treatment”.

Pp 3, lines 22-24. I do not understand what the authors mean when they say “This is due to its facilitation of consensus and consistency among many observers regarding their findings”. How can AI facilitate “consensus and consistency” among a group of pathologists, for example? First, they would really have to trust the AI system to align their diagnoses with the AI’s diagnoses, and as we all know, trust in AI has been an issue in several medical disciplines. Secondly, how can AI affect the pathologists’ intra-observer agreement, so as to improve their “consistency”? Please clarify.

Pp 3, lines 24-26. Furthermore, the authors claim that “Artificial intelligence is essential for assessing breast cancer and quantifying mitotic cells”. Who has determined that AI is “essential” for these tasks? Pathologists have been doing these tasks for a long time, with no AI, and achieving good results, so I do not see how one can claim AI to be “essential” in this task. Please tone down your enthusiasm.

Pp 3, line 29. Which are the “previous methods” that the authors are referring to? Please be specific regarding what you are talking about.

Introduction

History of Artificial Intelligence

Pp 4, lines 14-40. I still don’t see what is the purpose of this sub-section, as this paper is not about AI, but about AI’s application in breast cancer diagnoses.

Concepts in Artificial Intelligence

Pp 4, line 46. Typo: It should read “AI applications in Medicine have evolved…”.

Machine Learning (ML)

Pp 5, lines 5-30. Fine.

Deep Learning (DL)

Pp 5, line 57. It should read “Deep Learning is a subset of Machine Learning…”.

Artificial Intelligence in Medicine

Pp 6, lines 27-43. Fine.

Artificial Intelligence in Cancer Diagnosis

Pp 6, line 51. I believe that it is too premature to say that “Artificial intelligence is significantly advancing cancer diagnosis” when in fact most AI models are not deployed in the clinic yet. In this way, I believe that saying that “AI has the potential to significantly advance cancer diagnosis” is a better representation of the current state of events.

Pp 7, lines 17-20. Again, it seems premature to say “AI technology is improving the accuracy of clinical image analysis for identifying cancer progression, aiding in early detection and diagnosis…”. I think that because a technology has the potential to make improvements to a certain task, it does not mean that it is making improvements to that task. I would suggest that the authors be just a bit more careful when making statements like these.

Artificial Intelligence in Breast Cancer Pathology

Pp 8, lines 38-40. When the authors say that the size of the histopathology images are “100k x 100k”, which unit of measure are they referring to? Pixels? Please clarify.

Pp 9, lines 17-38. The authors present a number of AI models that have been developed to perform at different tasks, and their performances. What is missing from this presentation is a statement indicating that all of these algorithms were developed and evaluated in a laboratory setting, and that none of them has actually been deployed in the clinical practice. I think that this important point is not explicitly stated, and readers may get the impression that all of these algorithms are being used in the clinic.

Pp 10, lines 6-8. Can the authors please explain what they mean when they say “significantly reducing the time required for pathologists to read slides tumors”. What are “slides tumors”?

Methodology

Fine as written.

Results

Literature Search and Screening

Pp 13, lines 27-32. The authors say that, from the 3113 studies potentially found, after applying the inclusion criteria and filtering out the duplicates, 1194 unique studies were selected. From these, 1516 studies were excluded. I am confused. How can one exclude 1516 studies out of a group of 1194 studies? Please clarify.

Characteristics of the Included Studies

Pp 14, line 41. Instead of “histopathology pictures”, please use “histopathology images”.

Pp 17-24. It’s a bit overwhelming to have all 5 Tables being presented to the reader one after the other, without any context. I would suggest presenting each table as it is called in the text, so as to give the reader a reference for what is displayed in the Table.

Summary of Findings

Mitotic Index Assessment and Quantification

Pp 26, lines 10 and 24. I would not call a model developed in 2018 a “A recently developed deep learning model”. Please correct.

Discussion

Pp 28, line 35. When the authors say that “Commercially available AI models specifically designed for breast cancer exist”, do they mean for breast cancer “diagnosis”, for breast cancer “detection”? Please clarify.

Pp 28, lines 49-51. The authors state that “Nonetheless, various limitations persist in obstructing its extensive implementation on a larger scale”. However, this was never discussed in the Results section, even though it is an important piece of information to understand why, despite all of AI’s prowess (as highlighted by the authors) it has not reached large distribution in the clinical practice. Can the authors please include a section on the “obstacles that currently prevent AI’s extensive implementation on a larger scale” in their Results section?

Pp 30, line 53. I don’t believe that the authors should state that “Similar to our findings about AI’s performance in tissue classification…”. The issue here is that there was no experiment conducted in this study, and as such, there were no “findings” per se. It would be best if instead the authors used “Similar to the reported findings about AI’s performance in tissue classification…”

Pp 30-31, lines 55 and 3. Similarly, it is not appropriate for the authors to say “Our data indicate that AI excelled in both molecular subtyping and Ki67 computation”. The authors didn’t run any experiments that produced any “data” to indicate that. Instead, it would be best to say “The data from the studies reported in this review indicate that AI excelled…”

Pp 31, lines 17-22. It is a circular argument to state that “Artificial Intelligence enhanced inter-pathologist agreement rates […] when aided by a pre-trained AI assistant tool”. Please remove the end of the sentence, as it is already clear from the start of the sentence what the authors mean to say.

Pp 31, line 43. The authors name a sub-section “Other obstacles to widespread adoption of AI”. Given that they have discussed no obstacles thus far, why use the word “Other” in this sub-section heading?

Pp 31-32, lines 52-6. I do not agree with the statement made by the authors that “A primary procedural drawback of most contemporary models is their need on extensive, annotated datasets for training. Manual annotation is labor-intensive and exhibits variability both among and across pathologists, undermining the fundamental objective of the AI models”. Only supervised (or semi-supervised) learning algorithms require “extensive, annotated datasets for training”. Unsupervised learning algorithms do not require any labelled data for training. As for stating the supervised algorithms comprise “most contemporary models”, I am not sure about that. I think the authors should rethink this statement, or at the very least provide references to support what they are saying.

Pp 32, line 10. The authors say that “… impact the efficacy of AI models in rare tumour categories”. But unless one is very naïve or unfamiliar with AI models, in reality nobody would expect AI to do well in rare tumor categories, exactly for the reason that the authors mention, namely, lack of training samples. So while this is certainly a barrier, I do not believe it to be one of the primary barriers for AI’s clinical adoption.

Pp 32, lines 19-21. The authors say that “Consequently, each AI tool necessitates validation and verification under these specific conditions”. However, are they aware that, at least in the US, once an AI model receives FDA certification, the model is locked, that is, it cannot learn new things? In this way, once an FDA certified model is acquired by a healthcare system, there is no way to adapt the model to the characteristics of the local population, and if that population is significantly different from the population on which the model was trained, the model is not going to perform well, and if the vendor wants to incorporate data from this new population into the training of the algorithm they will have to re-certify the AI model? This is also a significant barrier for adoption of AI in the clinical practice.

Conclusions and Future Directions

Pp 33, line 17. The authors start by saying that “Various AI tools can…” I think that it is too early to say that they “can”, given all of the obstacles described by the authors. I believe that a better word to use here would be “may”.

---

## [Editor Report]

I thank the authors for their effort in revising the work. I agree with the suggestions of the reviewer and they will greatly improve the manuscript. Hence I recommend a simple minor revision.

---

## [Editor Report]

Dear Authors,

thank you for revising the manuscript. The paper has improved, and I recommend that it be accepted.